# Maternal Depression during Pregnancy and Postpartum Period among the Association of Southeast Asian Nations (ASEAN) Countries: A Scoping Review

**DOI:** 10.3390/ijerph20065023

**Published:** 2023-03-12

**Authors:** Seo Ah Hong, Doungjai Buntup

**Affiliations:** ASEAN Institute for Health Development, Mahidol University, Nakhon Pathom 73170, Thailand; seoah.hon@mahidol.ac.th

**Keywords:** antenatal depression, postnatal depression, prevalence, ASEAN

## Abstract

Identification of mothers with depression is important because untreated perinatal depression can have both short- and long-term consequences for the mother, the child, and the family. This review attempts to identify the prevalence of antenatal and postnatal depression (AD and PD, respectively) of mothers among the ASEAN member countries. A literature review was conducted using PubMed, Scopus, and the Asian Citation Index. The reviews covered publications in peer-reviewed journals written in the English language between January 2010 and December 2020. Of the 280 articles identified, a total of 37 peer-reviewed articles conducted in 8 out of 11 ASEAN member countries were included. The Edinburgh Postnatal Depression Scale (EPDS) was the most common instrument used to identify depression. This study showed the number of studies reporting the prevalence of AD was 18 in five countries. For PD, 24 studies in eight countries were included. The prevalence of AD ranged from 4.9% to 46.8%, and that of PD ranged from 4.4% to 57.7%. This first review among ASEAN countries showed very few studies conducted in lower-middle-income and substantial heterogeneity in prevalence among studies reviewed. Further research should be conducted to estimate the prevalence using a large representative sample with a validated assessment tool among the ASEAN countries.

## 1. Introduction

Pregnancy and the postpartum period can be joyful times as well as times of stress and difficulties, since delivery and childcare bring several physiological and psychosocial changes for mothers [1,2]. Consequently, they are the periods of increased vulnerability for the onset or relapse of a mental illness [3]. The untreated depression during pregnancy persists to the postpartum period [4]. The perinatal depression further results in obstetric outcomes and birth complications (e.g., low birth weight and pre-term birth) [5,6] and poor physical health and cognition of the infant [7,8,9]. Perinatal depression is thus a considerable public health problem affecting the mother, her baby, and her family [8,10].

The prevalence of antenatal and postnatal depression (AD and PD, respectively) is highly variable between populations and across cultures [11,12,13,14]. The prevalence of maternal mental disorders is reported to be greater in low- and middle-income countries (LMICs) than in high-income countries [5,15]. A recent meta-analysis in LMICs showed that about 25.3% (95% CI: 21.4–29.6%) and about 19.0% (15.5–23.0%) of women experienced AD and PD, respectively [16]. Despite this, mothers at risk during pregnancy or delivery often go undetected and underdiagnosed in LMICs [6,17,18], since the maternal depression is given less priority for intervention due to the belief that it does not immediately cause fatalities.

The Association of Southeast Asian Nations (ASEAN) is a regional intergovernmental organization that promotes economic, political, and security cooperation among its 11 countries in Southeast Asia [8], including Timor-Leste, which joined in 2019. In line with the United Nation’s Sustainable Development Goals (SDGs) [19], mental health is identified as one of the health priorities under the ASEAN Post-2015 Health Development Agenda for 2016–2020 [20]. Nonetheless, according to the Mental Health Report of 2016, many of the member states are faced with several challenges, such as a lack of resources to fund mental health services and community-based interventions and a mental health workforce shortage. Consequently, maternal depression during pregnancy or the postpartum period still remain neglected in many of the ASEAN member states, with no standard specific guidelines for maternal mental healthcare services. Although some studies of the ASEAN member countries were included in previous international and regional reviews [6,12,14,16], to our knowledge, there is a lack of information specific to the region. Moreover, the ASEAN countries have a great diversity in terms of socioeconomic conditions ranging from lower-middle-income (e.g., Cambodia, Lao PDR, Myanmar, the Philippines, Timor-Leste, and Vietnam) to high-income countries (e.g., Brunei Darussalam and Singapore), as well as culture and religion, which may contribute to differences in the prevalence of perinatal depression. This highlighted the need to identify recent epidemiological evidence of the prevalence of AD and PD among the ASEAN countries. The identification of the scope of existing studies and comprehensive understanding of the phenomena of interest may increase awareness of the need for timely and appropriate mental health interventions and programs among the vulnerable population in ASEAN countries [20], which has so far been neglected, and offer insights into how well-prepared ASEAN countries are for achieving the SDGs.

## 2. Methods and Materials

The primary focus of the review is on prevalence of maternal depression during pregnancy to the postpartum period. Descriptive research methodology was used to review peer-reviewed literature. A literature review was conducted to gather data from various sources and to ensure proper understanding of the research subject. The Preferred Reporting Items for Systematic Reviews and Meta-Analysis (PRISMA) chart (Figure 1) shows the phases of paper identification and selection.

The main inclusion factor was peer-review scientific journals indexed in selected electronic databases, such as Scopus, PubMed, and the Asian Citation Index (ACI) databases. Articles were searched using several combinations of search words and their synonyms with the Boolean operators “AND” or “OR”, searching titles, keywords, and abstracts (Table 1). The language selected was English. Studies published during the last 10 years (i.e., between January 2010 and December 2020) were included because they were more likely to reflect the current state of knowledge on maternal depression from pregnancy to the postpartum period.

Furthermore, articles were selected based on the inclusion and exclusion criteria (Table 1). The exclusion criteria were as follows: editorials, letters, intervention studies, interviews, no access to full text since the methodological quality of the papers could not be assessed, and no assessment of prevalence, a primary focus of the study. Studies on high-risk populations (some population groups are at considerably higher risk of developing AD and PD than others, such as women living with HIV, type 2 diabetes mellitus, preterm labor, caesarean section, and psychiatric problems (women with known psychiatric diagnosis such as schizophrenia or anxiety disorders)) were excluded because the risks of these populations developing AD and PD are high. Furthermore, studies conducted outside ASEAN member countries were excluded, since the findings of these studies recruiting Asian minorities as part of their samples might be different from the general population in each country because they were possibly influenced and complicated by other cultural and environmental factors.

## 3. Results

### 3.1. Literature Search

Studies were identified using Scopus, PubMed, and ACI. The database search with our selected keywords returned a total of 280 records (Figure 1). Of those, 100 records’ titles and abstracts were reviewed. After review, 45 articles were eligible for full-text review. Methodological issues, such as study design, primary focus of study, and target population were assessed by the authors, and discussions were conducted to resolve any discrepancies in decisions about excluding or including articles. After the stepwise elimination, 37 records were included in the final review. The total number of included studies for the literature review by country are shown in Table 2. A total of 37 peer-reviewed articles conducted in 8 out of 11 ASEAN member countries were included in this review, namely Indonesia (n = 2), Lao PDR (n = 1), Malaysia (n = 8), Philippines (n = 1), Singapore (n = 7), Thailand (n = 6), Timor-Leste (n = 2), and Vietnam (n = 10).

The Edinburgh Postnatal Depression Scale (EPDS) was the most common instrument used to identify AD and PD (29 out of 37 studies). Other screening tools included were the Center for Epidemiological Studies—Depression (CES-D), the Hospital Anxiety and Depression Scale (HADS), the Mini International Neuropsychiatric Interview (MINI), the self-reported questionnaire (SRQ), the Structured Clinical Interview for Diagnostic and Statistical Manual of Mental Disorders (DSM-IV) Axis I Disorders (SCID-I), and Whooley questions.

### 3.2. Characteristics, Measurement, and Prevalence of AD

The number of studies on prevalence of AD (including four longitudinal studies presenting both AD and PD) was 18, namely studies in Malaysia (n = 5), Singapore (n = 4), Thailand (n = 3), Timor-Leste (n = 2), and Vietnam (n = 4) (Table 2). As shown in Table 3 and Table 4, of the 18 studies, the majority of the studies collected data longitudinally (n = 10) and in hospitals or health clinics (n = 15) and during the third trimester of pregnancy (n = 7). The most common instrument used to identify AD was the EPDS (n = 14) and the most used cut-off scores of the EPDS to screen for AD were varied between countries and among studies of each country. For example, the most used cut-off scores of the EPDS were 12 for Malaysia, 15 for Singapore, 13 for Timor-Leste, and 10 for Vietnam. The prevalence of AD ranged from 4.9% in Vietnam to 46.8% in Thailand. Eight studies reported a prevalence of 5.0–14.9%, while eight studies reported a prevalence of 15.0–25.0%, one study reported a prevalence of less than 5%, and one study reported a prevalence greater than 25.0%. By country, the prevalence ranged from 10.3% to 20.0% in Malaysia, 7.1% to 17.0% in Singapore, 19.3% to 19.7% in Timor-Leste, and 4.9% to 24.5% in Vietnam (Figure 2).

### 3.3. Characteristics, Measurement, and Prevalence of PD

The number of studies reporting the prevalence of PD was 24 in eight member countries. The studies were conducted in Indonesia (n = 2), Lao PDR (n = 1), Malaysia (n = 3), Philippines (n = 1), Singapore (n = 4), Thailand (n = 4), Timor-Leste (n = 1), and Vietnam (n = 8) (Table 2). As shown in Table 4 and Table 5, the majority of the studies collected data cross-sectionally (n = 12), up to 4 months postpartum (n = 16), and at hospitals (n = 14). Although the assessment tools used were different between studies, the most common instrument used to identify PD was EPDS (n = 18). The different cut-off scores used between countries made the direct comparison difficult between studies. The prevalence ranged from 4.4% in Malaysia to 41.7% in Thailand (Figure 3). One study reported less than 5% and five studies more than 25.0% prevalence. Eleven studies reported prevalence of 5.0–14.9%, while seven studies reported prevalence of 15.0–25.0%. By country, the prevalence ranged from 19.9% to 26.2% in Indonesia, 31.8% in Lao PDR, 4.4% to 14.3% in Malaysia, 16.4% in Philippines, 9.0% to 23.3% in Singapore, 5.3% to 41.7% in Thailand, 12.6% in Timor-Leste, and 8.2% to 27.6% in Vietnam (Figure 3).

## 4. Discussion

To identify the needs for maternal mental healthcare, evidence on the prevalence of maternal perinatal depression is important. This scoping review represents the first attempt to identify the burden of perinatal depression (e.g., AD and PD) in the 11 ASEAN member countries. This study showed substantial variation in prevalence of AD and PD between countries with very few studies conducted in lower-middle-income countries and between studies due in part to variations in study design and methods of assessment. Further research should be conducted to provide the best information on mothers at risk of perinatal depression aimed at addressing maternal mental problems at national and regional levels particularly in resource-poor settings.

Our study revealed substantial difference in number of studies on prevalence of perinatal depression among the ASEAN countries. The included 37 peer-reviewed articles for AD and/or PD were from 8 out of 11 countries. Many of the studies included were from Vietnam, followed by Malaysia, Singapore, and Thailand. On the other hand, few studies were available in the remaining member states, specifically Lao PDR, Philippines, and Timor-Leste, and no studies in Brunei, Cambodia, and Myanmar met the inclusion criteria. The very limited number of studies in those countries, except Brunei, can be associated with the high burden of maternal mortality. Maternal and child health (MCH) is given top priority in policy and strategy development among several ASEAN members with high maternal mortality rate: 121 deaths per 100,000 livebirths for Philippines and 142 for Timor-Leste, 160 for Cambodia, 177 for Indonesia, 185 for Lao PDR, and 250 for Myanmar [20]. It appears that most health policies in these countries focus on issues causing high mortalities, and thus minimal attention is paid to the maternal depression during pregnancy and postpartum periods. It should be highlighted that the neglected issue of maternal mental health during the periods may consequently contribute to failure in the great reductions of maternal and child mortality and morbidity [58], since untreated symptoms of maternal depression can have negative consequences for the health of women and their neonates and infants [33,59].

This study showed that AD and PD were prevalent but ranged widely across the studies in the ASEAN countries. Regarding AD, 18 studies in five countries were included in this review. The majority ranged from 5.0% to 25.0%: Eight studies reported a prevalence of 5.0–14.9%, while eight studies reported 15.0–25.0%. Meanwhile, more studies on PD were conducted (25 studies in eight countries), and the prevalence seems wider. One study reported less than 5%, and 11 studies reported prevalence of 5.0–14.9%, while seven studies reported prevalence of 15.0–25.0%, and five studies reported more than 25.0% prevalence. More attention has been drawn to the presence of depression among women during the postpartum period compared to antenatal period. This may be explained by many new challenges, such as caring for a new infant and also the serious negative impact on the mother and their family as well as child developmental outcome. Childcare with insufficient family and social support can lead to poor sleep [60,61] and anxiety and stress due to the mother’s need to take care of herself and baby, including feeding practices [39,40,49,53,62,63,64], which further lead to PD. Furthermore, the wider range of prevalence of PD compared to AD may be associated with negative obstetric/pediatric factors, such as history of caesarean section, abnormal signs of the fetus and infant, pregnancy complications, and maternal ill health. Poor obstetric/pediatric outcomes, prevalent in many LMICs, may result in higher prevalence of PD in line with previous studies [5,15]. This indicates that healthcare services should introduce psychological support in the treatment plan for women who have obstetric complications from pregnancy, particularly in LMICs. However, due to the limited number of studies in lower-middle-income countries, such as Cambodia, Lao PDR, Myanmar, the Philippines, and Timor-Leste, we cannot say that the prevalence of PD is high in lower-middle-income countries. Rather, a pattern of decreased prevalence rates was found in studies using either longitudinal study design or cross-sectional study design with a nationally representative sample. The substantial difference in the prevalence of AD and PD in our study may be due in part to the study design used and methods of assessment. The included studies on AD and/or PD were either cross-sectional or longitudinal studies, which are useful for reporting prevalence. While about 60% of the included studies on AD used a longitudinal approach, the majority of the studies on PD used cross-sectional study design with small sample sizes. Among cross-sectional studies, only two studies estimated the prevalence using a nationwide representative sample in Thailand [50] and Malaysia [43]. This may suggest that more efforts are required to identify the accurate estimation of the burden using a nationwide representative sample among pregnant and postnatal mothers, and there is a clear need for rigorously designed longitudinal studies to improve our understanding of perinatal depression in ASEAN countries. In addition, although the EPDS was identified as the most frequently used measurement tool in the ASEAN region, the cut-off points used to define the perinatal depression and timing in the assessment of the problem resulted in the wide disparities in the reported prevalence estimates across the included studies. For example, the cut-off scores of EPDS used were varied: either ≥10 or ≥11 for AD and ranging from ≥11 to ≥13 for PD in Thailand and ≥10 to ≥13 for AD and PD in Vietnam. Therefore, further study on validation of the EPDS using a larger sample representing the overall population of perinatal women, particularly in lower-middle-income countries where no or few validation studies have been conducted, should be followed. Nonetheless, given the substantial prevalence of maternal depression during pregnancy and after birth, this review provides valuable insights into the importance of timely detection of maternal depression during pregnancy and after birth and the development of policy and programs aimed at addressing issues which have so far been neglected in the ASEAN countries.

The WHO [65] recommends integrating mental health services into primary care, and this has also been applied to LMICs, as there are no evidence-based guidelines for LMICs. Skilled healthcare professionals, including midwives, nurses, and physicians under strong supervision, monitoring and supportive structures would also be necessary to identify the occurrence and severity of maternal perinatal depression through timely detection and to deliver essential psychosocial care and mental health support to pregnant and postpartum mothers at the primary care level in resource-limited settings [15,58,66]. However, mental health services in most of the ASEAN member countries have not been integrated into the primary healthcare system due to human resource shortage (either few or no mental health specialists), while some countries, such as Singapore, Brunei Darussalam, and Thailand, have implemented integrated mental health services into primary care levels as well as hospitals [67]. Furthermore, there is little evidence available on how to integrate evidence-based interventions into non-specialized care settings, such as primary care, maternal and child healthcare, and community care settings in LMICs. The recommended antenatal care visits (four or more times) varied considerably but were relatively low in most countries in the ASEAN region (60–70% in Timor-Leste and Myanmar to 80% in Cambodia, and around 90% in Philippines and Indonesia) [68]. Postnatal care visits are much less frequent than those of antenatal care across the countries due to the majority of women in the lower-middle-income countries delivering at home. Despite the existence of a mental health policy plan in the ASEAN region, most of the ASEAN member countries had no plans including maternal mental health as warranting special attention or focus.

Findings must be interpreted with caution due to some weaknesses of this review. First, the review includes studies written in English for international comparison and indexed in the three chosen databases. Although additional literature searches were conducted in Google and Google Scholar to determine whether articles were missing from the three chosen databases, we may have missed some original articles for ASEAN member countries. In addition, most of the studies are subject to selection and measurement bias which occur by use of a cross-sectional study design, small non-representative samples, convenience sampling, and variability in the instrument measures and scale, which contribute to heterogeneity in results and hinder the generalization of the results. Next, we did not perform a meta-analysis of the prevalence. However, the findings drawn from the review may be useful to understanding the burden of perinatal depression of mothers and to developing planned maternal mental healthcare in the ASEAN countries by providing an overview of maternal perinatal depression across the countries.

## 5. Conclusions

The findings of this review showed that published research is still not sufficient, particularly in lower-middle-income countries of the ASEAN members. The substantial variation in prevalence of AD and PD between countries and between studies is due in part to variations in study design and methods of assessment. Further research should be conducted to provide the best information on mothers at risk of perinatal depression aimed at addressing maternal mental problems at national and regional levels in the ASEAN member countries.

## Figures and Tables

**Figure 1 ijerph-20-05023-f001:**
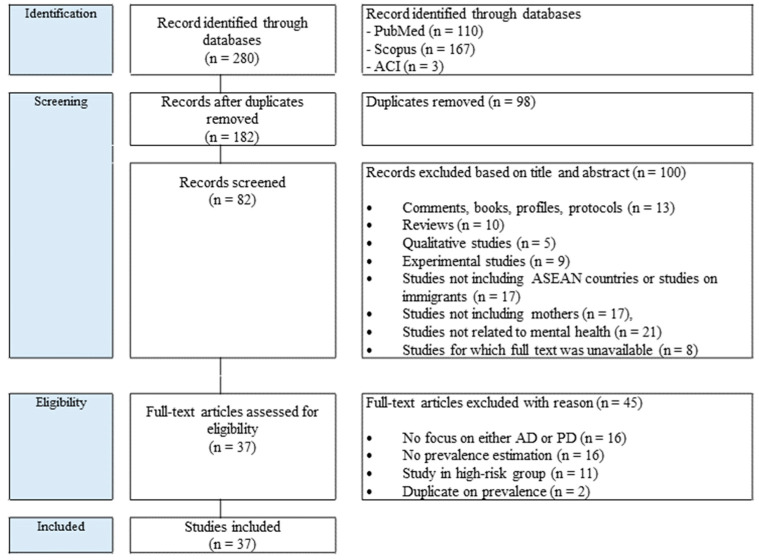
PRISMA diagram.

**Figure 2 ijerph-20-05023-f002:**
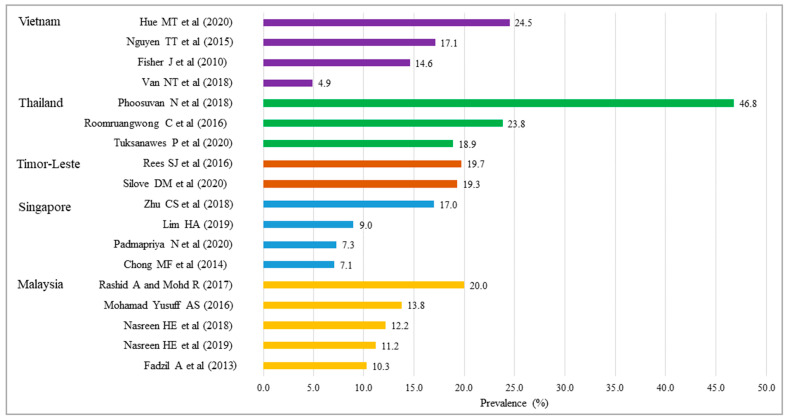
Prevalence of antenatal depression in the ASEAN member countries [21,22,23,24,25,26,27,28,29,30,31,32,33,34,35,36,37,38].

**Figure 3 ijerph-20-05023-f003:**
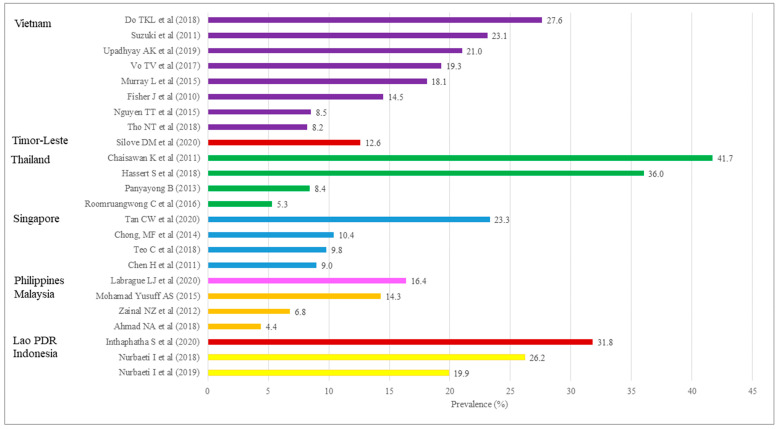
Prevalence of postpartum depression in the ASEAN member countries [34,35,36,37,38,39,40,41,42,43,44,45,46,47,48,49,50,51,52,53,54,55,56,57].

**Table 1 ijerph-20-05023-t001:** Search strategies.

Database	Scopus, PubMed, ACI
Keywords	“depressive symptoms” OR “Depression”; “Antenatal” OR “Postnatal” OR “Postpartum” OR “Perinatal”; “Thailand” OR “Vietnam” OR “Cambodia” OR “Laos PDR” OR “Malaysia” OR “Philippines”, OR “Singapore” OR “Myanmar” OR “Indonesia” OR “Brunei” OR “Timor-Leste”. Combined terms: 1 AND 2 AND 3.
Inclusion criteria	Peer-reviewed articles published from Jan 2010–Jan 2021, studies that report prevalence of antenatal and postpartum depression following childbirth, and English publications.
Exclusion criteria	The following studies were excluded because they were not the focus of the study. - Review papers; - Studies related to psychiatric problems; - Studies that were conducted among high-risk groups of women, such as women living with HIV and type 2 diabetes mellitus, experiencing preterm labor and caesarean section, women facing threatened miscarriage, and refugees and migrants; - Studies with descriptive statistics to determine the associated factors; - Studies determining the impact of depression or validation studies; - Studies reporting neither prevalence nor determinants.

**Table 2 ijerph-20-05023-t002:** Number of studies by outcome measures.

	New Country Classification Income ^(1)^	By Outcome Measures ^(2)^			Total
		AD	Both AD and PD	PD	
Brunei Darussalam	High	0	0	0	0
Cambodia	Lower-middle	0	0	0	0
Indonesia	Upper-middle	0	0	2	2
Lao PDR	Lower-middle	0	0	1	1
Malaysia	Upper-middle	5	0	3	8
Myanmar	Lower-middle	0	0	0	0
Philippines	Lower-middle	0	0	1	1
Singapore	High	3	1	3	7
Thailand	Upper-middle	2	1	3	6
Timor-Leste	Lower-middle	1	1	0	2
Vietnam	Lower-middle	2	2	6	10
Total		13	5	19	37

(1) The new country income classifications for the World Bank’s 2020 fiscal year. (2) When there are two studies determining factors within a different framework but reporting the same prevalence using the same cohort data, we included both studies but counted as one study for the studies reporting prevalence.

**Table 3 ijerph-20-05023-t003:** Prevalence of antenatal and postnatal depression and their determinants.

Country	Authors (Year) (Ref)	Study Design	Study Setting	Time Frame	Sample Size	Instruments	Prevalence of AD
Malaysia (n = 5)	Fadzil A et al. (2013) [21]	Cross-sectional	Hospital	Any stage of pregnancy	175	M-HADS (≥8); MINI	10.3%; 8.6%
	Rashid A and Mohd R (2017) [22]	Cross-sectional	Health clinics	Any stage of pregnancy	3000	M-EPDS (≥12)	20.0%
	Mohamad Yusuff, A. S. (2016) [23]	Prospective cohort	Health clinics	36–38 weeks	2072	M-EPDS (≥12)	13.8%
	Nasreen HE et al. (2018) [24]	Prospective cohort	Health clinics	Third trimester	904	M-EPDS (≥12)	12.2%
	Nasreen HE et al. (2019) [25]	Prospective cohort	Health clinics	At third trimester of pregnancy and at birth	799	M-EPDS (≥12)	East coast: 11.2%; West coast: 13.6%
Singapore (n = 3)	Lim HA et al. (2019) [26]	Longitudinal	Hospital	Up to 14 weeks	926	EPDS (≥15)	9.0%
	Padmapriya N et al. (2016) [27]	Longitudinal	Two major hospitals	26–28 weeks	1144	EPDS (≥15)	7.3%
	Zhu CS et al. (2018) [28]	Cross-sectional	Hospital	5–12 weeks + 6 days	241 Women—uncomplicated pregnancy	EPDS (≥13)	17.0%
Thailand (n = 2)	Phoosuwan N et al. (2018) [29]	Cross-sectional	Community and provincial hospitals	28–37 week	449	T-EPDS (≥10)	46.8%
	Tuksanawes P et al. (2020) [30]	Cross-sectional	Hospital	Any stage of pregnancy	402	T-CES-D (≥19)	18.9%
Timor-Leste (n = 1)	Rees SJ et al. (2016) [31]	Cross-sectional	Antenatal clinics	3–6 months of pregnancy	1672	EPDS (>13)	19.7%
Vietnam (n = 2)	Hue MT et al. (2020) [32]	Cross-sectional	Four hospitals	Any stage of pregnancy	1260	EPDS (≥10)	24.5%
	Van NT et al. (2018) [33]	Prospective cohort	Hospitals	Up to 24 weeks	1276	EPDS (≥10)	4.9%

The Edinburgh Postnatal Depression Scale (EPDS); the initial before EPDS indicates the country’s initial and validated local language versions of the EPDS. For example, T-EPDS indicates validated Thai language versions of the EPDS. Hospital Anxiety and Depression Scale (HADS); Mini International Neuropsychiatric Interview (MINI).

**Table 4 ijerph-20-05023-t004:** Prevalence of perinatal (both antenatal and postnatal) depression.

Country	Authors (Year) (Ref)	Study Design	Study Setting	Time Frame	Sample Size	Instruments	Prevalence of AD and PD
Singapore (n = 1)	Chong MF, Wong et al. (2014) [34]	Prospective cohort study (GUSTO)	Two large birthing hospitals	Time 1: 26–28 weeks’ gestation Time 2: 3 months postpartum	Time 1: n = 967 Time 2: n = 719	EPDS (≥15 for AD; ≥13 for PD)	AD = 7.1% PD = 10.4
Thailand (n = 1)	Roomruangwong C et al. (2016) [35]	Longitudinal	Hospital	Time 1: third trimester pregnancy; Time 2: 2–3-day follow-up; Time 3: 4–6 weeks after delivery	126	T-EPDS (≥11)	AD = 23.8%; PD = 7.8% at Time 2 and 5.3% at Time 3
Timor-Leste (n = 1)	Silove DM et al. (2020) [36]	Longitudinal	Four community health centers	Time 1: third-6th month of pregnancy Time 2: followed up 18 months postpartum	1292	EPDS (≥13)	AD = 19.3%; PD = 12.6%
Vietnam (n = 2)	Nguyen TT et al. (2015) [37]	Population-based prospective	Communities	Either the last trimester of pregnancy or 4–6 weeks after giving birth and followed up 15 months later	234	SCID-I	PeD = 17.1%; PD = 8.5%
	Fisher J et al. (2010) [38]	Cross-sectional	Community health centers	-At least 7 months pregnant -4th to 8th week postpartum	Pregnant women (n = 199) Mothers of young infants (n = 165)	SCID	AD = 14.6%; PD = 14.5%

Antenatal depression (AD); perinatal depression (PeD); postnatal depression (PD). The Edinburgh Postnatal Depression Scale (EPDS); the initial before EPDS indicates the country’s initial and validated local language versions of the EPDS. For example, T-EPDS indicates validated Thai language versions of the EPDS. The Structured Clinical Interview for DSM-IV Axis I Disorders (SCID-I).

**Table 5 ijerph-20-05023-t005:** Prevalence of postnatal depression and their determinants.

Country	Authors (Year) (Ref)	Study Design	Study Setting	Time Frame	Sample Size	Instruments	Prevalence of PD
Indonesia (n = 2)	Nurbaeti I et al. (2019) [39]	Cross-sectional	Two public health centers	Postpartum	166	EPDS (≥12)	19.9%
	Nurbaeti I et al. (2018) [40]	Prospective longitudinal	Public health centers	1, 2 and 3 months postpartum	283	EPDS (≥13)	18.37% at 1 month; 15.19% at 2 months; 26.15% at 3 months
Lao (n = 1)	Inthaphatha, S et al. (2020) [41]	Cross-sectional	Four central hospitals	6–8 weeks postpartum	428	EPDS (≥10)	31.8%
Malaysia (n = 3)	Zainal NZ et al. (2012) [42]	Cross-sectional	Hospital	After 4 weeks post-delivery	411	MINI	6.8%
	Ahmad NA et al. (2018) [43]	Nationwide cross-sectional	Government primary care clinics	6–16 weeks postpartum	5727	M-EPDS (≥12)	4.4%
	Mohamad Yusuff, AS et al. (2015) [44]	Prospective cohort	Health clinics	At 1, 3 and 6 months postpartum	2072	M-EPDS (≥12)	7.1% at 1 month 6.9% at 3 months 7.6% at 6 months 14.3% during the first 6 months
Philippines (n = 1)	Labrague LJ et al. (2020) [45]	Cross-sectional	Five maternal facilities	6 weeks postpartum	165	EPDS (≥10)	16.4%
Singapore (n = 3)	Tan CW et al. (2020) [46]	Longitudinal	Two major hospitals	3 months postpartum	651	EPDS (≥10)	23.3%
	Chen H, Wang J et al. (2011) [47]	Prospective cohort (KKH)	Two major hospitals	2 weeks to 6 months postpartum	1367	EPDS (≥13)	9.0%
	Teo C et al. (2018) [48]	Prospective cohort study (GUSTO)	Two large birthing hospitals	2: 3 months postpartum	490	EPDS (≥13)	9.8%
Thailand (n = 3)	Chaisawan K et al. (2011) [49]	Cross-sectional	Hospitals	6-weeks postpartum	84 (teenage mothers)	T-CES-D (>16)	41.7%
	Panyayong B (2013) [50]	Nationwide cross-sectional	Provincial hospitals	6–8 weeks after delivery	1731	T-EPDS (≥13)	8.4%
	Hassert S et al. (2018) [51]	Cross-sectional	Hospitals	Up to 12 months postpartum	161	EPDS (≥12)	36.0%
Vietnam (n = 6)	Suzuki et al. (2011) [52]	Cross-sectional	Hospitals	1–3 months postpartum	299	Whooley Questions (2 questions)	23.1%
	Murray L et al. (2015) [53]	Cross-sectional	Community Health Centers	1–6 months postpartum	431	EPDS (≥13)	18.1%
	Vo TV et al. (2017) [54]	Cross-sectional	Communities	4 weeks to 6 months postpartum	600	EPDS (≥13)	19.3%
	Do TKL et al. (2018) [55]	Cross-sectional	Communities	Up to 1 year postpartum	116	EPDS (≥12)	27.6%
	Upadhyay AK et al. (2019) [56]	Longitudinal (Young Lives Study; YLS)	Communities	5–21 months postpartum	1835	SRQ	21.0%
	Tho NT et al. (2018, 2019) [57]	Prospective cohort (PAVE)	Hospitals and community health stations	4–12 weeks postpartum	1274	EPDS (≥10)	8.2%

The Center for Epidemiological Studies—Depression (CES-D). The Edinburgh Postnatal Depression Scale (EPDS); the initial before EPDS indicates the country’s initial and validated local language versions of the EPDS. For example, T-EPDS indicates validated Thai language version of the EPDS. The Mini International Neuropsychiatric Interview (MINI). Self-reported questionnaire (SRQ).

## Data Availability

Not applicable.

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
