# Peer review of "Maternal Depression during Pregnancy and Postpartum Period among the Association of Southeast Asian Nations (ASEAN) Countries: A Scoping Review"

_ijerph, 2023, doi:10.3390/ijerph20065023_

Round 1

Reviewer 1 Report

Comments and suggestions are attached in the pdf file below.

Reviewer 2 Report

This study is meaningful in that it reviewed the prevalence of depression of mothers among the ASEAN countries. 

  • There is a lack of sufficient explanation as to why we should look at the prevalence of depression of mothers among the ASEAN countries. Please supplement the purpose and necessity of the study.

  • There is a large difference in economic level between the countries included in the study (for example, Singapore and Myanmar), and a close examination is needed to see if maternal mental health is ignored even in a country like Singapore.

  • Since only papers written in English are included, it is necessary to review whether there is a high possibility of existing studies written in the language of each country

  • As a result of this study, it was suggested that perinatal depression was high. What is the criterion for high?

  • Discussion section needs to contain lots of rich arguments, which is very worthwhile. The points would read better with a clear that what are the summaries of the main findings, what research implications and suggestions for future research are discussed, and what kinds of health care implications can be made.

Round 2

Reviewer 1 Report

The authors have revised the manuscript in the right way. The process for publication check should be continued.